# Golyadkin Confronts Perceptual Distance and Curvatures: Coping with Ambiguity

## Abstract

The adversarial vulnerability of classifiers reveals a core divergence: ML systems make distinctions without difference; biological systems tolerate difference without distinction– and survive because of it.

Adversarial vulnerability is analyzed through decision boundaries and distance-based perturbation models. However, the distances used do not match true perceptual distances and the overall approach fails to account for the misalignments with perceptual topology and geometry. We discuss contexts in which the perceptual distance is computable. In particular, we discuss image recognition contexts in which the perceptual distance between any two inputs is finite. The finiteness underpins an inherent, and informally accepted by the ML community, vulnerability of classifiers defined on such images, rendering all labels susceptible to adversarial attacks.

This demonstrates why some valiant attempts to achieve robustness may be doomed. And yet, biological systems function and thrive despite or may be even because of the ever-present ambiguity. Systems function not because they are robust but because they are sufficiently conceptually coherent. The notions of coherency, conceptual coherency, coherency failure rate, and the conceptual margin of a labeled data set are defined and discussed in this paper.

We define latent adversarial vulnerability, showing that vulnerability arises not only from adversarial perturbations but also through conceptual drift along perceptual Sorites, and introduce perceptual curvature which can be used to identify latent adversarial vulnerability regions.

## 1 Introduction

The vulnerability of classifiers to adversarial examples – pairs of imperceptibly different inputs that are assigned conflicting class labels – has spawned copious flood of research efforts to eliminate it.[1] Identifying a perceptually meaningful distance, to measure imperceptibility and perceptual difference between inputs/stimuli is considered essential for quantifying adversarial vulnerability and for adversarial training. Until recently it was accepted that true perceptual distance cannot be easily defined and computed. Instead, various approximations have been proposed. There is a substantial empirical evidence that these distances do not provide good approximation of the true perceptual distance, Sharif et al. (2018); Laidlaw et al. (2021); Sen et al. (2020); Ghildyal and Liu (2023). Recognizing this mismatch between the proposed approximations and human perception Croce et al. (2025) proposed imperceptibility measures based on CLIP models and tested them on measuring perceptual change. However, there is no evidence that they approximate human performance on measuring perceptual difference. Detecting and measuring perceptual change differs fundamentally from detecting and measuring perceptual difference. The former involves memorization and memory recall, and comparing a stored mental image with a stimulus, whereas the later is a direct comparison

---

[1]The definition of adversarial examples has been somewhat of a moving target, initially set to be "imperceptibly small perturbations to a correctly classified" input (Szegedy et al. (2013)) to examples that are perceptually different but are "designed to cause a mistake", i.e., to mis-classify, Elsayed et al. (2018).

between stimuli. In Section 3 we discuss the true perceptual distance and its computational operationalization.

While robust classifiers are keenly pursued, there has long been an informal understanding within the ML community that truly robust classifiers may not always be attainable. Indeed, unless the classification is cast as a **well defined classification problem** adversarial vulnerabilities cannot be entirely eliminated, Kamberov (2024). In Section 3.1 we discus fundamental problems which are not well defined. Yet, biological systems appear to perform classification even in contexts when robust classifiers do not exist. We propose that biological systems have evolved to be pragmatic. Their classifications are conceptually coherent at best or at least sufficiently coherent.

In Section 3.2 we formally define the notions **coherent classifiers**, **conceptually coherent** classifiers, **coherency failure rate**, the **conceptual margin** of a labeled data set, and **latent adversarial vulnerability**– a previously unrecognized class of adversarial vulnerabilities emerging through perceptual drift and requiring a paradigm shift in adversarial defense strategies. We further show that – when indiscriminability and the encounter probability are suitably aligned – labeled data sets with sufficiently high conceptual margin may be used to train classifiers that are both perfectly accurate and conceptually coherent.

In Section 3.3 we introduce perceptual curvature which can be used to flag areas in the space of inputs **X** where latent adversarial vulnerability might lurk.

## 2 Doppelgängers, Regular Classifiers, and All That . . .

Perception defines a context-relevant topology $\tau_{\tilde{\delta}}$ on the space of inputs **X**. Adversarial Doppelgängers occur precisely when a **perceptual tile** - a connected component of the perceptual topological space $(\mathbf{X}, \tau_{\tilde{\delta}})$—is intersected by multiple classifier regions, Kamberov (2024; 2025).

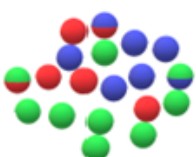

Figure 1: The perceptual topological space $(\mathbf{X}, \tau_{\tilde{\delta}})$ is the disjoint union of perceptual tiles, represented as balls; the classifier regions $\{R_1, R_2, R_3\}$ are colored red, green, and blue, respectively. Adversarial Doppelgängers live on the tiles that intersect more than one classifier region.

The context-relevant perceptual topology $\tau_{\tilde{\delta}}$ is defined by the pre-basis formed by the **phenomenal neighborhoods** $\{\mathfrak{d}(x)\}_{x \in \mathbf{X}}$:

$$\mathfrak{d}(x) = \left\{ y \in \mathbf{X} \,:\, y \overset{\alpha\tilde{\delta}}{\approx} x \right\}{}^2 \tag{1}$$

where $\overset{\alpha\tilde{\delta}}{\approx}$ is the Williamson **indiscriminability relation** – commonly referred to as the perceptually indistinguishable relation in ML papers. The perceptual tiles are precisely the equivalence classes defined by the transitive closure $\sim_\sigma$ of the indiscriminability relation. [3]

Our lived experience, robust empirical observations, and extended epistemological debates all lead to the observation that perceptual topology is rarely – if ever – metric, and even

---

[2]The Doppelgängers of an input $x$ are the other members of its phenomenal neighborhood: $\mathfrak{d}(x) \setminus \{x\}$. They are ubiquitous in **X**, a part of a mechanism evolved to manage uncertainty and to reduce cognitive stress.

[3]$x \sim_\sigma y$ iff $x$ and $y$ can be connected by a chain of Doppelgängers, $x \overset{\alpha\tilde{\delta}}{\approx} x_1 \overset{\alpha\tilde{\delta}}{\approx} \cdots \overset{\alpha\tilde{\delta}}{\approx} x_n \overset{\alpha\tilde{\delta}}{\approx} y$.

more unlikely manifold topology. The core of the explanation is simple: separability axioms fail in real contexts.

Still, humans, and likely other organisms, employ context-relevant perceptual distance $d_\infty$ to quantify the perceptual similarity between inputs. The perceptual distance is an extended distance function $d_\infty : \mathbf{X} \times \mathbf{X} \to [0, +\infty]$ defined as the graph distance on a graph whose vertices correspond to the inputs $x \in \mathbf{X}$ and an edge $\{x, y\}$ between the vertices $x, y \in \mathbf{X}$ exists if and only if $x \overset{\alpha\delta}{\approx} y$, Kamberov (2024). The resulting extended distance – also referred to as the **degrees of separation** – is perceptually grounded measure of similarity even when computationally convenient classical and novel deep learning networks-based measures are not. The distance between different perceptual tiles is infinite, while the distance between, inputs that belong to the same tile is finite, and so the perceptual tiling of $\mathbf{X}$ is referred to as an *infinitely separated perceptual tiling.* **The extended perceptual metric does not generate the perceptual topology, but it recovers indiscriminability as a limiting case of similarity.**

Adversarial Doppelgängers robust classifiers are, in fact, the perceptually regular classifiers introduced in Kamberov (2024; 2025).[4] The existence of a regular classifier $\Omega = \{\Omega_1, \ldots, \Omega_m\}$ is equivalent to the existence of a **perceptually coherent class partition** of the space of inputs/stimuli $\mathbf{X}$.[5] We are not aware of any ML models that are perceptually coherent. However, a fundamental goal in ML is to build and train classifiers that match a perceptually coherent class partition. The classifier **conceptual accuracy** accuracy$_\Omega(R)$ provides a measure of the mismatch between a classifier $R = \{R_1, \ldots, R_m\}$ and a perceptually coherent class partition $\Omega = \{\Omega_1, \ldots, \Omega_m\}$. The classifier conceptual accuracy is defined as:

$$\text{accuracy}_\Omega(R) = \sum_{i=1}^m \mu(R_i \cap \Omega_i). \tag{2}$$

In practice, latent coherent partitions may be difficult to identify and characterize analytically and operationally. Instead, the ML community works with finite sets of inputs $S = \{x_1, \ldots, x_M\} \subset \mathbf{X}$ and labeled datasets, $L(S) = \{(x_1, l_1), (x_2, l_2), \ldots, (x_M, l_M)\} \subset \mathbf{X} \times \{1, 2, \ldots, m\}$. The **observed testing accuracy** of a classifier $R$ on a labeled data set $L(S)$ is defined as the fraction of labeled data points in $L(S)$ which are correctly classified by the classifier $R$:

$$a(R; L(S)) = \frac{1}{M} \#\{(x_i, l_i) \in L(S) : \text{label}_R(x_i) = l_i\} \tag{3}$$

Maximizing the observed testing accuracy on benchmark datasets remains a central objective in ML. It is widely accepted that high testing accuracy does not guarantee alignment with a perceptually coherent partition. The pursuit of 'robust' classifiers – that is, classifiers which are not vulnerable to adversarial attacks is often motivated by this premise.

In Section 3.2 we show that rather than searching for high accuracy robust classifiers, given any labeled data set, it is meaningful to identify classifiers that achieve high accuracy while maintaining adversarial vulnerability below the maximal tolerable threshold, $\epsilon < 1$, specific to the given context, everywhere except on rare inputs.

## 3 Perceptual Distance, Coherence, and Curvature

At first glance, operationalizing the computation of $d_\infty(x, y)$ appears daunting. Yet, even single-cell organisms routinely compute small perceptual distances. Humans often perform distance computations in parallel and in multiple modality-specific brain regions, which suggests that perceptual distance computation constitutes a canonical neural process, as defined in Carandini and Heeger (2012). The structure of the neural substrates for canonical neural computations must ensure speedy processing and reliability through redundancy while maintaining a minimal footprint to support parallel execution of multiple instances.

---

[4]The name *regular* comes from elliptic regularity theory, since their labeling functions are harmonic with respect to a specific perceptual Laplace operator, Kamberov (2024).

[5]Perceptually coherence means that $(x \in \Omega_i) \Rightarrow (y \in \Omega_i), \forall y \overset{\alpha\delta}{\approx} x$.

The foundational investigations in psychophysics by Weber and Fechner provide insight in the perceptual topology of the space of inputs $\mathbf{X} = (0, +\infty)$ which in turn is used to represent essential inputs/stimuli including visual and audio inputs.

**Example 0:** Let $\mathbf{X} = (0, +\infty)$. Suppose that Weber's law holds and let $k > 0$ be the Weber constant and $w = 1 + k$. Let

$$\mathfrak{d}(x) = (x/w, xw) \tag{4}$$

The covering $D_{\alpha\delta} = \{\mathfrak{d}(x)\}_{x \in \mathbf{X}}$ defines a perceptual indiscriminability relation, $\overset{\alpha\delta}{\approx}$, on $\mathbf{X}$, s.t., $x \overset{\alpha\delta}{\approx} y$ if and only if $x, y \in \mathfrak{d}(x) \cap \mathfrak{d}(y)$, and a perceptual topology $\tau_\delta$ as the topology generated by the sub-basis $\mathfrak{D}_{\alpha\delta} = \{\mathfrak{d}(x)\}_{x \in \mathbf{X}}$. There exists a neural network whose input is a pair of stimuli $(x, y) \in \mathbf{X} \times \mathbf{X}$ which "computes" whether $x \overset{\alpha\delta}{\approx} y$. It is gated and has hard activation functions. The network and the weights are shown in Figure 2.

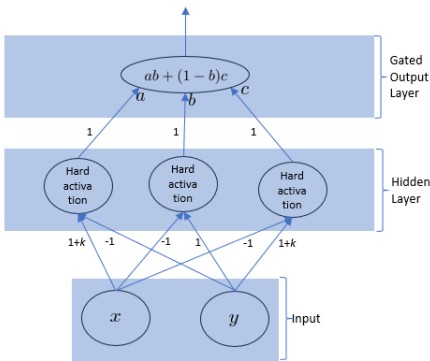

Figure 2: Perceptual discrimination neural substrate: returns 1 if $x \overset{\alpha\delta}{\approx} y$, otherwise it returns 0.

The hard activation functions are all equal to

$$\mathrm{h}(z) = \begin{cases} 0, & z \le 0 \\ 1, & z > 0. \end{cases} \tag{5}$$

This model also provides a tool to estimate perceptual distance by deciding whether $d_\infty(x, y) \le 1$. In conjunction with a network which decides whether $x = y$ these neural substrates enable the computation of small perceptual distances (i.e., distances less than 2).[6] Perceptual discrimination and the estimation of small perceptual distances are canonical neural computations. Correspondingly, the neural model in Figure 2 is remarkably – and emphatically – not a deep learning model. It is narrow, shallow, and fundamentally nonlinear – a model shaped by evolutionary pressures, optimized for speed and redundancy, enabling versatile applications.

Interestingly, by replacing the weights $1 + k$ with $(1+k)^n$ in the discrimination model, Figure 2, we obtain a model that decides whether $d_\infty(x, y) \le n$ for $n = 2, 3, \dots$. In particular, the evolutionary critical perceptual discrimination and the computation of small perceptual distances are supported by optimized neural substrates – while the computation of larger distances requires additional resources, including combining multiple neural systems.

The model processes stimuli in different modalities including audio, visual, olfactory, and tactile inputs in different contexts and aligns with well documented brain plasticity phenomena Citri and Malenka (2008); Lövdén et al. (2010); Carcagno and Plack (2011); Wenger and Kühn (2021).

---

[6]While it is easy to find an artificial neural network which decides an input $x$ is identical to another input $y$, the ability of humans to establish identity is a long debated subject in epistemology and the study of perception and cognition.

### 3.1 When no Label is Safe: Drifting into Adversity.

Many classifiers report extraordinary performance on specific classification problems. However, the classification problems are not well defined – the underlying classes are ambiguous – and so none of these classifiers are, and in fact cannot be, adversarially robust. This conceptual ambiguity is a fundamental issue and has been known to philosophers and cognitive psychologists and linguists Rosch (1973); Dummett (1975); Lakoff (1987). Hue and intensity are likely the most extensively studied ambiguous perceptual inputs[7] and have drawn sustained attention as canonical examples where perceptually coherent (unambiguous) concepts do not exist and hence adversarially robust classifiers are unattainable.

**Example 1:** Let $\mathbf{X}$ be a space of all flat color gray scale images where each image has a single uniform color, and the only distinguishing factor between any two images is the difference in intensity. Assuming that Weber–Fechner's observations hold: namely, that perceptual discrimination thresholds scale logarithmically with stimulus intensity, the perceptual distance between any two images is finite. In fact one can construct a non-stationary random walk connecting any image $x_0 \in \mathbf{X}$ with an image $x_w$ that is perceptually indistinguishable from a solid "white image", i.e., image of maximum intensity explicitly illustrating that the topological space $(\mathbf{X}, \tau_\delta)$ is connected, $\mathbf{X}$ has a single perceptual tile. The walk is driven by a random walk in perceptual intensity values

$$I_n = I_{n-1} + \xi_n, \quad \xi_n \sim U(a_n, b_n), n \geq 1 \tag{6}$$

where $\xi_n$ is the random increment at step $n$, and $U(a_n, b_n)$ represents the uniform distribution in the range $[a_n, b_n]$. The variability of perceptual sensitivity is encoded in the evolution of the interval $[a_n, b_n]$ and results in the non-stationarity of the walk.

**Example 2:** Let $\mathbf{X}$ be a space of $h$-by-$w$ gray scale images containing the MNIST data set. Assuming that Weber-Fechner's observations/law hold, the perceptual distance between any two images $x_0, x_M \in \mathbf{X}$ is finite. Thus a finite chain of Doppelgängers can be constructed as a $hw$-dimensional extension of the walk defined in Example 1. As any classifier $R$ – such that $\mathrm{label}_R(x_0) \neq \mathrm{label}_R(x_M)$– traverses this chain, it inevitably encounters adversarial examples, as illustrated in Figure 3.

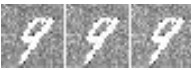

Figure 3: A short (3-step) non-stationary walk of Doppelgängers. The first and the second images are perceptually indistinguishible, as are the second and third. CoPilot assigns three different labels to the images: *Known* (clearly interpretable content, e.g., handwritten "g") to the first image, *Unknown* (impossible to interpret) to the second (middle) image, and *CAPTCHA* (contains distorted text typical of CAPTCHA challenges) to the third image.

This happens not because the *model R fails*, it happens because *no label is safe*.

### 3.2 Pragmatism and Coherency

Robustness is an attractive ideal and has been established as a classifier design objective. Yet, perceptual ambiguity renders it unattainable in many important contexts, despite numerous valiant efforts in adversarial training.

You cannot fix what isn't there.

From a pragmatic and survival standpoint, what truly matters is that the likelihood of encountering conceptual ambiguity is negligible or at least acceptably low.

A classifier $R$ is called $\delta$-$\epsilon$ **coherent** if

$$\mu\left(\{x : \mu(\mathfrak{d}(x; R)) > \delta\}\right) \leq \epsilon \tag{7}$$

---

[7]Though sometimes ambiguity is conflated with vagueness – the inability to verbally draw boundaries that nevertheless exist and are perceived Kennedy (2007).

where

$$\mathfrak{d}(x; R) = \{y \in \mathfrak{d}(x) : \text{label}_R(y) \neq \text{label}_R(x)\} \tag{8}$$

is the collection of $R$-adversarial Doppelgängers of $x$ – those inputs perceptually indistinguishible from $x$ but labeled differently by $R$.

An $\delta$-$\delta$ coherent classifier is called $\delta$ **coherent**[8] and **conceptually coherent** if $\delta = \epsilon = 0$, i.e.,

$$\mu\left(\{x : \mu(\mathfrak{d}(x; R)) > 0\}\right) = 0. \tag{9}$$

We define the **coherency failure rate** of a classifier $R$, as

$$0 \leq \epsilon(R) = \inf\left\{\epsilon \geq 0 \,:\, \mu\left(\{x \,:\, \mu(\mathfrak{d}(x; R)) > \epsilon\}\right) \leq \epsilon\right\} \leq 1. \tag{10}$$

The coherency failure rate of a conceptually coherent classifier is zero. Conceptually coherent classifiers are practically regular – or, equivalently, practically robust: the probability to encounter an adversarial Doppelgänger, in the specific task context, is negligible. Conceptual coherence does not rule out highly vulnerable inputs; however, these are operationally inaccessible in the task context. On the other hand, the probability to find an adversarial $R$-Doppelgänger to any specific accessible input is negligible.

Crucially – and happily – conceptually coherent classifiers can still exist, even when the classification problem is not well defined. We will show an example below (Example 5). In practice, we may not even need a conceptually coherent classifier, but rather one whose coherency failure rate does not exceed a tolerable threshold.

**Example 3:** Let $\mu(A) = \frac{2\sqrt{\pi}}{\pi} \int_A e^{-t^2} \, dt$ be the probability measure on $\mathbf{X} = (0, +\infty)$ and let the indiscriminability relation on $\mathbf{X}$ be defined by the covering $\mathfrak{D}_{\alpha\delta} = \{\mathfrak{d}(x) = (x/w, xw)\}_{x \in \mathbf{X}}$, where $w > 1$ is a fixed constant. Let $\gamma > 0$ and let $R(\gamma)$ be the linear classifier defined by

$$\text{label}_{R(\gamma)}(x) = \begin{cases} 1, & 0 < x < \gamma \\ 2, & \gamma \leq x. \end{cases} \tag{11}$$

A direct computation shows that every classifier $R(\gamma)$ is $(\text{erf}(w\gamma) - \text{erf}(\gamma/w))$ coherent. In fact, the set of inputs whose AD vulnerability exceeds $(\text{erf}(w\gamma) - \text{erf}(\gamma/w))$ is negligible, every $R(\gamma)$ is $(\text{erf}(w\gamma) - \text{erf}(\gamma/w)) - 0$ coherent.

While neither robust nor conceptually coherent binary classifiers exist in this context, for every fixed threshold $\tau > 0$, there exists a continuum of binary classifiers whose coherency failure rate is below the threshold $\tau$.

The adversarial vulnerability of classifiers is commonly – and often loosely – attributed to generalization failure. In this paper we define the relevant concepts. The **perceptual generalization** of a set of inputs $S \subset \mathbf{X}$ is the union of phenomenal neighborhoods

$$\mathbf{X}_\sigma(S) = \bigcup_{x \in S} \mathfrak{d}(x) \subset \mathbf{X} \tag{12}$$

For a finite set of labeled data

$$L(S) = \{(x_1, l_1), (x_2, l_2), \ldots, (x_M, l_M)\} \in \mathbf{X} \times \{1, 2, \ldots, m\} \tag{13}$$

we define the **conceptual margin** $\rho_\infty(L(S))$ as

$$\rho_\infty(L(S)) = \min\left\{d_\infty(x_i, x_j) \,:\, l_i \neq l_j\right\} \tag{14}$$

and call the labeled set a **perceptually regular labeled data set** if its conceptual margin is strictly bigger than one,

$$(l_i \neq l_j) \Rightarrow (d_\infty(x_i, x_j) > 1).^9 \tag{15}$$

---

[8]Clearly, if $\delta \geq \epsilon$, then every $\delta$-$\epsilon$ coherent classifier is $\delta$-coherent. Every classifier is $\delta$-$\epsilon$ coherent for some $0 \leq \delta, \epsilon \leq 1$. The $\epsilon$ provides a bound on the probability to find inputs that are at least $\delta$-vulnerable.

[9]It is not known if any of the state of the art labeled datasets used in practice are regular.

Furthermore, we will denote by $G(L(S))$ the set of all classifiers $R = \{R_1, \ldots, R_m\}$ which have testing accuracy one, i.e., $\mathrm{label}_R(x_i) = l_i$ for every $i = 1, \ldots, M$, and such that

$$\mathrm{label}_R|_{\mathfrak{d}(x_i)} \equiv l_i, \forall (x_i, l_i) \in L(S).^{10} \tag{16}$$

The classifiers that belong to $G(L(S))$ have testing accuracy one on $L(S)$ and **generalize perceptually at least to order one**, $\mathrm{label}_R(y) = \mathrm{label}_R(x_i) = l_i$ for every $y \overset{\alpha\delta}{\approx} x_i$.

If the labeled data set is not perceptually regular, that is, if there exist inputs $x_i, x_j \in \mathbf{X}$ such that $x_i \overset{\alpha\delta}{\approx} x_j$ and $\mathrm{label}_R(x_i) \neq \mathrm{label}_R(x_j)$, then $G(L(S)) = \emptyset$.

Classifiers trained on perceptually regular labeled datasets with low conceptual margins can suffer from subtle, previously unknown adversarial vulnerabilities.

**Example 4:** Let $R$ be a classifier with a sampling accuracy equal to one on the perceptually regular labeled dataset $L(S)$, i.e., $a(R; L(S)) = 1$. Suppose that $R$ has certifiable robustness radius equal to one at $x_i \in S$, meaning $\mathrm{label}_R(y) = \mathrm{label}_R(x_i), \forall y \in \mathfrak{d}(x_i)$. If there exists a data point $(x_j, l_j) \in S$, such that $d_\infty(x_i, x_j) = 2$ and $\mathrm{label}_R(x_j) = l_j \neq l_i = \mathrm{label}_R(x_i)^{11}$, then there exists an adversarial attack $\varepsilon(x_j)$ such that $x_i \overset{\alpha\delta}{\approx} \varepsilon(x_j) \overset{\alpha\delta}{\approx} x_j$, with $\mathrm{label}_R(\varepsilon(x_j)) = l_i \neq l_j = \mathrm{label}_R(x_j)$. Adversarial training to eliminate the vulnerability at $x_j$ will result in a classifier that is 'safe' at $x_j$; in fact, it will have a certified robustness radius one at that point. However, this process introduces adversarial Doppelgängers vulnerability at the previously safe point $x_i$ and possibly many other inputs $y \in \mathfrak{d}(x)$. In this scenario, suppressing one vulnerability inevitably creates others elsewhere.

This phenomenon, which we term a **latent adversarial vulnerability**, can be exploited by an attacker either to directly target specific, seemingly safe inputs or to impose a potentially crippling resource strain through a loop of attacks – what we refer to as **Ouroboros attacks**. Latent adversarial vulnerability is insidious but not accidental – it is related to the geometry of perceptual space discussed in Section 3.3.

If the conceptual margin of $L(S)$ is greater or equal to three, $\rho_\infty(L(S)) \geq 3$, then $G(L(S)) \neq \emptyset$ and if the conceptual margin of the labeled set $L(S)$ is at least four degrees of separation, there are useful bounds on the size of the adversarial vulnerability, $\mu(\mathfrak{d}(x; R))$, at each input $x \in \mathbf{X}_\sigma(S)$, for every classifier $R \in G(L(S))$.

**Observation 1.** *Given the data set $S = \{x_1, \ldots, x_M\}$ and the labeled dataset $L(S) = \{(x_1, l_1), (x_2, l_2), \ldots, (x_M, l_M)\} \in \mathbf{X} \times \{1, 2, \ldots, m\}$ such that*

$$\rho_\infty(L(S)) > 3. \tag{17}$$

*Let $R = \{R_1, \ldots, R_m\} \in G(L(S))$, then every $y$ which is an $R$-adversarial Doppelgänger of an input $x \in \mathbf{X}_\sigma(S) \setminus S$, belongs to $\mathbf{X} \setminus \mathbf{X}_\sigma(S)$ and therefore, $R$ is $(1 - \mu(\mathbf{X}_\sigma(S)))$-coherent.*

**Proof:** Indeed, if $x \in \mathbf{X}_\sigma(S) \setminus S$ the existence of an $y \in \mathfrak{d}(x; R) \cap \mathbf{X}_\sigma(S)$ would imply the existence of a short perceptual Sorites chain $x_j \overset{\alpha\delta}{\approx} y \overset{\alpha\delta}{\approx} x \overset{\alpha\delta}{\approx} x_i$, where

$$l_j = \mathrm{label}_R(x_j) = \mathrm{label}_R(y) \neq \mathrm{label}_R(x) = \mathrm{label}_R(x_i) = l_i. \tag{18}$$

Thus $d_\infty(x_i, x_j) \leq 3$ and $l_i \neq l_j$, which is ruled out by the Equation (17) and since $\mathfrak{d}(x_i; R) = \emptyset$, we obtain the bound $0 \leq \mu(\mathfrak{d}(x; R)) \leq 1 - \mu(\mathbf{X}_\sigma(S)), \forall x \in \mathbf{X}_\sigma(S)$ and therefore, $R$ is $(1 - \mu(\mathbf{X}_\sigma(S)))$-coherent. $\square$

More generally adopting the notation from Sossinsky (1986), we define the $n$**-fold perceptual thickening** of a set of inputs $S \subset \mathbf{X}$ as

$$\mathbf{X}_{n\sigma}(S) = \{y \; : \; d_\infty(y, S) \leq n\}. \tag{19}$$

Furthermore, we will denote by $G_n(L(S))$ the set of all classifiers $R = \{R_1, \ldots, R_m\}$ which have testing accuracy one, i.e., $\mathrm{label}_R(x_i) = l_i$ for every $i = 1, \ldots, M$, and such that

$$\mathrm{label}_R|_{\mathbf{X}_{n\sigma}(\{x_i\})} \equiv l_i, \forall (x_i, l_i) \in L(S).^{12} \tag{20}$$

---

[10]Thus the certified perceptual robustness radius at each data point $x_i \in S$ is at least one.

[11]Hence, $\rho_\infty(L(S)) \leq 2$.

[12]Thus the certified perceptual robustness radius at each data point $x_i \in S$ is at least $n$.

The classifiers that belong to $G_n(L(S))$ have testing accuracy one on $L(S)$ and **generalize perceptually at least to order** $n$.[13]

Using the triangle inequality we obtain a generalization of Observation 1.

**Observation 2.** *Given the data set* $S = \{x_1, \ldots, x_M\}$ *and the labeled dataset* $L(S) = \{(x_1, l_1), (x_2, l_2), \ldots, (x_M, l_M)\} \in \mathbf{X} \times \{1, 2, \ldots, m\}$ *such that*

$$\rho_\infty (L(S)) > 2n + 1, \tag{21}$$

*then:* **(a.)** $G_n(L(S)) \neq \emptyset$*;* **(b.)** *If* $R = \{R_1, \ldots, R_m\} \in G_n(L(S))$*, every* $y$ *which is an R-adversarial Doppelgänger of an input* $x \in \mathbf{X}_{n\sigma}(S)$*, belongs to* $\mathbf{X} \setminus \mathbf{X}_{n\sigma}(S)$*.*

The following corollary shows that it is possible to bound effectively – and in some cases practically eliminate adversarial vulnerability by training on perceptually coherent data.

**Corollary 1.** *If* $\rho_\infty (L(S)) \geq 2n + 2$ *, then every* $R \in G_n(L(S))$ *is* $(1 - \mu(\mathbf{X}_{n\sigma}(S)))$*-coherent.*

**Example 5:** Let $\mathbf{X} = (0, +\infty)$ and let the indiscriminability relation on $\mathbf{X}$ be defined by the covering $\mathfrak{D}_{\alpha\delta} = \{\mathfrak{d}(x) = (x/w, xw)\}_{x \in \mathbf{X}}$, where $w > 1$ is a fixed constant. If the probability measure on $\mathbf{X}$ has a piecewise constant distribution function, say,

$$\phi(x) = \begin{cases} c_1, & 0 < a_1 < x < a_2 \\ c_2, & w^2 a_2 < b_1 < x < b_2 \\ 0, & \text{otherwise,} \end{cases} \tag{22}$$

where $0 < c_1, c_2$ and $b_2 > w^2 b_1$, then there exist labeled sets $L(S)$ with conceptual margin $\rho_\infty (L(S)) > 3$ and such that the perceptual generalization of $S$ has full measure, $\mu (\mathbf{X}_\sigma(S)) = 1$. For each of these labeled sets, there exists a continuum of conceptually coherent perfectly accurate linear classifiers $R(\gamma)$ defined by Equation 11.

### 3.3 Latent Vulnerability and Geometry

Example 4 is a perceptually disturbing illustration of latent adversarial vulnerability.

**Definition 1.** A classifier $R$ has a **latent adversarial vulnerability** of depth $n \geq 0$ at the input $x \in \mathbf{X}$ if $\text{label}_R(y) = \text{label}_R(x)$, $\forall y$, s.t., $d_\infty(x, y) \leq n$, but there exists an input $z$ at perceptual distance $n + 1$, $d_\infty(x, z) = n + 1$, s.t., $\text{label}_R(z) \neq \text{label}_R(x)$.

Latent adversarial vulnerabilities of depth one pack a potential perceptual and cognitive punch – they enable adversarial Doppelgänsger attacks on safe data where retraining – intended to enhance robustness – will inadvertently remove their defenses. Latent adversarial vulnerabilities with depth more than one are equally insidious and more subtle: attackers and the targets may remain perceptually similar to safe data but the existing methods to measure perceptual similarity make these vulnerabilities difficult to detect and defend against. However, the existing work on adversarial attacks and robustness reveals that state of the art classifiers have many latent adversarial vulnerabilities.

Latent adversarial vulnerabilities are directly linked to and located in areas in the space of inputs $\mathbf{X}$ where there is **perceptual drift**, that is there are perceptual Sorites chains, $x_0 \overset{\alpha\delta}{\approx} \cdots \overset{\alpha\delta}{\approx} x_n$ with $x_n \notin \mathfrak{d}(x_0)$. The vulnerability is caused by the failure of a classifier to recognize and tolerate this perceptual drift; in effect the classifier's conceptual drift is misaligned with humans' ability to tolerate perceptual drift. It is imperative to be able to understand the shape of the perceptual terrain in order to locate sources of drift and the regions that support latent adversarial vulnerability. While the perceptual topology is rarely – if ever – a manifold topology or even metric topology we can still define curvature which can be used to identify locations where perceptual drift is possible. These 'perceptual' curvatures are close but different from the manifold and graph curvatures we know, because they have to work in non-separable, non-metric spaces and still flag locations where perceptual drift may emerge.

---

[13]Note, $\mathbf{X}_\sigma(S) = \mathbf{X}_{1\sigma}(S)$ and $G(L(S)) = G_1(L(S))$.

We define the **perceptual Ricci curvature** between the Doppelgängers $x \overset{\alpha\hat{\delta}}{\approx} y$ as

$$\Upsilon_\sigma\left((x,y)\right) = \mu(\mathfrak{d}(x))\mu(\mathfrak{d}(y) \setminus \mathfrak{d}(x)) + \mu(\mathfrak{d}(y))\mu(\mathfrak{d}(x) \setminus \mathfrak{d}(y)) \tag{23}$$

and the **perceptual curvature** at the input $x \in \mathbf{X}$ as

$$\Upsilon_\sigma\left(x\right) = \int_{\mathfrak{d}(x)} \Upsilon_\sigma\left((x,y)\right). \tag{24}$$

The perceptual Ricci curvature represents a field that separates the indiscriminable (perceptually indistinguishable) inputs.[14]

Perceptual Sorites chains emanating from $x$, do not exist if and only if $\mathfrak{d}(x) = \mathfrak{d}(y)$, $\forall y \in \mathfrak{d}(x)$ in which case $\Upsilon_\sigma\left(x\right) = 0$. Conversely, if $\Upsilon_\sigma\left(x\right) > 0$, then there exist perceptual Sorites chains trough $x$. Thus, we have the following operational observation:

**Observation 3.** *Positive perceptual curvature flags locations that present risk for latent adversarial vulnerability.*

In contrast with perceptual Ricci curvature, the perceptual curvature at an input appears to be deployable by at least humans, and possibly other organisms. Humans **know** whether there is a context-relevant perceptual chain including an input; this knowledge signals the computation of context-relevant perceptual curvature. For example, English speakers **know** that the term *crowd* can be embedded in a linguistic Sorites chain (e.g., *crowd → gathering → mob*). This awareness reflects the ability to deploy the knowledge that *crowd* has positive linguistic perceptual curvature while the term *one grain* has zero linguistic perceptual curvature. However, the *image of one grain* has positive visual perceptual curvature. Indeed, chains of images of grains have long served as the canonical example of Sorites chains.

## 4    SUMMARY AND DISCUSSION

The ML community has gradually become aware that in some contexts, adversarially robust classifiers are not achievable. This phenomenon is explained formally by perceptual ambiguity, Kamberov (2025), which is not a bug but an evolutionary developed mechanism enabling organisms to operate in complex environments. [15]

Yet, our living experience indicates that while robustness is desirable, sufficient coherency is often all that is possible – and all that is needed. The paper offers a formal approach to coherence and shows that sufficiently coherent and often practically coherent classifiers are achievable but that requires a careful selection of the training and benchmarking data. However, the conceptual margins of the existing benchmark data sets are not known. Until this is remedied, the development of sufficiently coherent classifiers and the perceptually valid benchmarking of classifiers are out of reach. An actionable program to compute conceptual margins is presented in the supplementary materials.

Latent adversarial vulnerabilities represent a hitherto unknown class of model failures. Crucially, they plague precisely classifiers that succeeded at becoming perceptually robust at some data points, and often at high cost. The perceptual curvature introduced in this paper can be used to alert about the risk of latent adversarial vulnerabilities. Humans routinely deploy knowledge of this curvature. However, to date we do not know how to 'code' these computations. As is the case with the perceptual distance, these may be canonical computations that require substrates that are different from today's large deep learning models. Developing such substrates remains an open challenge.

---

[14]The Perceptual Ricci curvature is not something that a perceptual system deploys to determine input distinctness, the inputs are, by definition, indistinguishable. Crucially, it differs from the Forman Ricci curvature, $F_\sigma\left((x,y)\right)$, which is defined on finite graphs Forman (2003) and can be extended to the perceptual graph, introduced in Kamberov (2024). Indeed, $\Upsilon_\sigma\left((x,y)\right) = \mu(\mathfrak{d}(x)) + \mu(\mathfrak{d}(y)) - F_\sigma\left((x,y)\right)$.

[15]This argument follows discussions involving *indiscernables*, Poincaré (1930) and *relevant differentials*, the fading from consciousness of *nearly constant . . . situations*, *consciousness* as a *phenomenon in the zone of evolution*, Schrödinger (1958).

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
