# OpenReview forum: "Golyadkin Confronts Perceptual Distance and Curvatures:  Coping with   Ambiguity"
_ICLR.cc/2026/Conference — ICLR 2026 Conference Withdrawn Submission_

### Official Review · Reviewer_ZYBz · 2025-10-30

**Soundness:** 2
**Presentation:** 2
**Contribution:** 2
**Rating:** 2
**Confidence:** 3

**Summary:**

The paper reframes adversarial robustness through perceptual topology: it models perception via an indiscriminability graph, defines an extended perceptual distance $d_\infty$ (finite within a “perceptual tile,” infinite across tiles), and argues that robustness often fails because ML decision regions cut across these tiles. It proposes replacing strict robustness with conceptual coherence, introducing metrics such as coherency failure rate and conceptual margin; when indiscriminability aligns with encounter probability and the margin is large enough, one can train classifiers that are perfectly accurate and coherent on the labeled set. The authors identify latent adversarial vulnerabilities that emerge through perceptual Sorites chains—even near points made safe by adversarial training—and introduce perceptual (Ricci) curvature to flag regions where such vulnerabilities lurk, illustrating ideas with grayscale/MNIST examples.

**Strengths:**

* Introduces a perceptual-topological lens on robustness (indiscriminability tiles, $d_\infty$, conceptual margin, coherence, perceptual curvature), offering a unifying vocabulary for why small perturbations can fool classifiers.
* Provides principled definitions and diagnostics that connect human indistinguishability to classifier behavior, pointing to testable conditions (e.g., margin thresholds) and dataset design levers.
* Motivations are intuitive (e.g., Sorites-style chains) and the conceptual framework is consistently developed, facilitating cross-talk with psychophysics and geometry.
* Reorients goals from absolute $\ell_p$-robustness to intra-tile conceptual coherence, with potential impact on benchmark construction, risk localization (via curvature), and future defense strategies.

**Weaknesses:**

*  No concrete algorithms or complexity analysis for estimating $d_\infty$, perceptual curvature, or conceptual margin from finite data.
*  Lacks systematic experiments (e.g., CIFAR/ImageNet) demonstrating diagnostics, risk localization, or gains over standard robust training.
*  Several key claims appear as observations without full theorems, assumptions, or proofs; boundary conditions and counterexamples are unclear.

**Questions:**

Can you report systematic results on CIFAR-10/100 (or an ImageNet subset)? Include: (1) correlations between your diagnostics (coherence failure, curvature, margin) and attack success/robust accuracy;  (2) comparisons and ablations vs. adversarial training.

---

> ### Author Response · Authors · 2025-12-04
>
> ZYBz: "No concrete algorithms or complexity analysis for estimating $d_\infty$, perceptual curvature, or conceptual margin from finite data."
>
> Factual corrections:
>
> (1)	The network in Example 0 is the canonical neural computation derived from Weber's Law, it is generalizable across all real sensory modalities that satisfy the law (hue, tone, brightness, etc.) as discussed in the paper it can be used to compute perceptual distances.
>
> (2)	An actionable method to compute perceptual margins and a example are included in the Supplementary materials which  are available for download on the OpenReview site.
>
> ZYBz:"Lacks systematic experiments (e.g., CIFAR/ImageNet) demonstrating diagnostics, risk localization, or gains over standard robust training."
>
> Factual corrections:
>
> (1)	Example 2 shows how one  can attack all classifiers trained on MNIST. We will provide the code that generates the non-stationary random walks needed to generate the attacks, c.f., Example 1.
>
> (2)	The role of perceptual curvature is precisely to flag regions of high risk.
>
> (3)	We are not aware of any SOTA algorithm that guarantees that the training data can’t be attacked by adversarial Doppelgangers. Observation 1 shows that if conceptual margin of a labeled data set is at least three ($\geq 3$), then we can construct classifiers such that the certified (perceptual) robustness radius at each training point is bigger or equal to one, i.e., the training data is not vulnerable to adversarial Doppelganger attacks.
>
> ZYBz: "Several key claims appear as observations without full theorems, assumptions, or proofs; boundary conditions and counterexamples are unclear".
>
> We have verified that the proofs of all observations are present. Some of the proof are very short but we will  edit the text to put the Proof environments.

---

### Official Review · Reviewer_zoRc · 2025-10-31

**Soundness:** 2
**Presentation:** 1
**Contribution:** 2
**Rating:** 2
**Confidence:** 3

**Summary:**

The paper raises a fair concern in the definition of adversarial examples, that is the pixel space is different from the perceptual space, and we need to develop methods and theoretical framework to measure distance in perceptual space instead of the pixel space to address this concern. The paper proposes the following contribution: (1) propose definition of "perceptual tile", "phenomenal neighborhood" based on Weber and Fechner indiscriminable measurement, (2) propose definition for delta-epsilon coherence measurement and coherence failure rate, (3) propose concept of conceptual margin and derive a bound to relate conceptual margin to the size of adversarial vulnerability. The paper mainly propose definitions, following with simple examples. No algorithm or empirical results are included.

**Strengths:**

The paper attempts to address a fair concern in the field of adversarial robustness, which is the adversarial robust needs to be measured on the perceptual space and not the perturbation in the pixel space. To address this concern, the authors propose a new measurement for robustness, which is "delta-epsilon coherence". The strength of the paper are:
1. The authors attempt to address a well-defined concern, which is methods to measure perceptual distance, and not pixel distance, and define the coherence of a classifier based on this measurement.
2. To my limited knowledge, the proposed concepts and definitions are novel in the adversarial robust community. And the proposed definition is precise and well-defined.
3. The authors demonstrate theoretically the importance of the "conceptual margin" in the dataset to define a theoretical bound on adversarial coherent (Eq. 17).

**Weaknesses:**

While the paper attempts to address a well-defined concern and proposed sound definition to measure coherence, the paper results are mainly based on illustrative examples and lack the significant results and comparison with current measurements and techniques to support their claims. While "delta-epsilon coherence" is a promising measurements to evaluate the classifier robustness. I would like to see either:
1. More detailed theoretical comparison of "delta-epsilon coherence" with current adversarial robustness measurement.
2. Empirical results on the application of this measurement on real dataset, and how they differ with the current adversarial robust measurements. L470 mentioned  "An actionable program to compute conceptual margins is presented in the supplementary materials.", but I didn't find any supplementary materials.

The paper also lacks the "Related Works" section that should include any previous works that try to attempt this problem.

Also, the paper's organization is quite difficult to follow (please see more details in the "Suggestion" below)

Suggestion:
1. The introduction section of the paper is a little bit confusing to see what is the paper outline and main contribution. To make it easier for readers to follow. I suggest:
	1. include an outline in the Introduction (for example, "the paper is structured as follow, in section 1, ..., in section 2, ..., etc...)
	2. Clearly list the contribution of the paper as a list (for example, "Here's our 3 main contribution")
2. In Section 2, the author list many different definitions and it's difficult to follow the relevance between them. I suggest to include a sentence that introduce all the definitions and how they relevant to each other (e.g, "In this section, we will introduce the following definitions:...")

**Questions:**

Question:
1. (Fig 2) Is `Fig.2` an original contribution of this neural circuit? Or it has appeared in other computational neuroscience papers?
2. (Section 3, L197) How does the network in `Fig.2` estimate the perceptual distance? I thought the Figure caption said that it would return a binary output for the `phenomenal neighborhood` ? How do we compute $d_{\infty}$ in general?
3. (Section 3, L299) Do we have a proof for this statement? "A direct computation shows that every classifier R(γ) is (erf(wγ) − erf(γ/w)) coherent"
4. (Section 4, L470) "An actionable program to compute conceptual margins is presented in the supplementary materials." -> Do we have the supplementary materials? I didn't see it in the main paper.
5. I thought the paper "Carandini and Heeger (2012)" is written about "normalization is a canonical neural computation", but the author cited that paper to claim "computing perceptual distance is a canonical neural computation"

---

> ### Author Response · Authors · 2025-12-04
>
> zoRc: (Fig 2) Is Fig.2 an original contribution of this neural circuit?
> Yes
> zoRc: (Section 3, L197) How does the network in Fig.2 estimate the perceptual distance? I thought the Figure caption said that it would return a binary output for the phenomenal neighborhood ? How do we compute  in general?
>
> We will update the test in the paper to show how this computation is done.
>
> zoRc: (Section 3, L299) Do we have a proof for this statement? "A direct computation shows that every classifier R(γ) is (erf(wγ) − erf(γ/w)) coherent"
>
> Yes.  We will update the test in the paper to show how this computation is done.
>
> zoRc: (Section 4, L470) "An actionable program to compute conceptual margins is presented in the supplementary materials." -> Do we have the supplementary materials? I didn't see it in the main paper.
>
> We confirm that the Supplementary materials zip file  949_Suppl.zip is uploaded on the OpenReview Site and managed to download it.
>
> zoRc: I thought the paper "Carandini and Heeger (2012)" is written about "normalization is a canonical neural computation", but the author cited that paper to claim "computing perceptual distance is a canonical neural computation"
>
> We thank the reviewer for this excellent question: Carandini and Heeger (2012) define canonical neural computations (CNC) and show that normalizations are CNC. Here we argue that since organisms perform perceptual discrimination continuously across multiple modalities, they and the equivalent computation of perceptual distances are CNC by deductive necessity.

---

### Official Review · Reviewer_kokB · 2025-10-31

**Soundness:** 3
**Presentation:** 2
**Contribution:** 2
**Rating:** 4
**Confidence:** 1

**Summary:**

Disclaimer: I am a computational neuroscientist by training, and have little to no knowledge of topology. I am thus not qualified to comment on the technical soundness of the arguments presented in this paper.

This submission explores properties of achievable classifiers that are defined on a particular type of topological space defined by the union of a set of "perceptual tiles."
Under one particular extended distance metric (the degree of separation on a graph where two points have an edge iff they are perceptually indistinguishable) points on a given particular perceptual tile have a finite distance between them whereas points on distinct perceptual tiles are infinitely far apart.
The author's provide an argument that some well studied stimulus spaces can be understood as having instantiations of this topology when using notions of indiscriminability from classic literature on perception.
A key result is that one consequence of this topology is that, if there are samples with different labels on a single perceptual tile, there will necessarily exist adversarial examples for any classifier.
Thus the author's argue that a more pragmatic goal to pursue is "pragmatism" or classifiers that accept the existence of adversarial examples for some set of rare instances rather than perfect robustness.
The paper goes on to describe settings where inducing defenses defenses to a particular vulnerability will always induce vulnerabilities elsewhere, and links this phenomenon ("latent vulnerabilities") and defines a curvature metric that can identify such vulnerabilities.

**Strengths:**

- Theoretical explorations of what computational properties are achievable for classifiers are valuable.
  - I.e. this paper proposes "shifting the goalpost" in the domain of robust learning from robustness everywhere to robustness with high probability almost everywhere. Which could be a useful paradigm shift.
- The documentation of a setting where inducing robustness at some data points (i.e. training points) will necessarily introduce vulnerabilities elsewhere is potentially very interesting (especially if this effect could be documented empirically).

**Weaknesses:**

- Some claims are stated without sufficient evidence:
  - Lines 40-41: "Until recently it was accepted that true perceptual distance cannot be easily defined and computed" I am not familiar with any such claim in the literature and would appreciate a citation. More generally this bit of exposition misses much of a large body of literature on perceptual metric learning. See for example LPIPS, MS-SSIM, DISTS papers.
  - Lines 111-115: "Still, humans, and likely other organisms, employ context-relevant perceptual distance d∞ to quantify the perceptual similarity between inputs" What evidence is there that this distance metric is particularly relevant for human perception? I did not see any quantitative comparison to human behavioral data in the cited paper (Kamberov, 2024)
  - Lines 152-154: What evidence is there that distance computation is a canonical computation in real neural circuits? This claim seems critically dependent on the one from 111-115, which I already feel is under supported. This is related to the claim on lines 199-200: "Perceptual discrimination and the estimation of small perceptual distances are canonical neural computations," what examples from the neuroscience literature support this claim?

- Some aspects of the presentation felt jarring to me. For example on line 99 the "Williamson indistinguishability relation" is introduced, but I did not know what this was and there was no citation provided. Similarly the introduction to Weber's law and the extension of Fechner felt insufficient.


- Nit: Line 228 "Assuming that Weber–Fechner’s observations hold: namely, that perceptual discrimination thresholds scale logarithmically with stimulus intensity". I think this is a mis-statement of Weber's law: discrimination thresholds scale linearly with intensity, and when this relation is integrated you can reach the conclusion that the perception of stimulus strength scales logarithmically with intensity.

**Questions:**

- It is unclear how to determine a priori if a classification problem is "coherent" in the language of this paper: i.e. is there a way in general to determine if there are examples of distinct classes within a single perceptual tile given access to a finite dataset?

- Is it possible to identify real systems for which latent vulnerabilities exist? In my opinion the strength of the contribution would be dramatically increased if (1) it could be shown in a practical system that inducing robustness around point A leads to decreased robustness around point B, and (2) that the perceptual Ricci curvature could be used to identify the risk of this type of vulnerability?

---

> ### Author Response · Authors · 2025-12-04
>
> kokB: Lines 40-41: "Until recently it was accepted that true perceptual distance cannot be easily defined and computed" I am not familiar with any such claim in the literature and would appreciate a citation. More generally this bit of exposition misses much of a large body of literature on perceptual metric learning. See for example LPIPS, MS-SSIM, DISTS papers.
>
> We discussed these metrics in the introduction. However, all of them are at best approximations of perceptual distance and as the references in the paper show they do not match human perception distance and similarity measures.
>
> kokB: Lines 111-115: "Still, humans, and likely other organisms, employ context-relevant perceptual distance d∞ to quantify the perceptual similarity between inputs" What evidence is there that this distance metric is particularly relevant for human perception? I did not see any quantitative comparison to human behavioral data in the cited paper (Kamberov, 2024)
>
> The very operation of perceptual discrimination is equivalent to compute perceptual distance.
>
> kokB: Lines 152-154: What evidence is there that distance computation is a canonical computation in real neural circuits? This claim seems critically dependent on the one from 111-115, which I already feel is under supported. This is related to the claim on lines 199-200: "Perceptual discrimination and the estimation of small perceptual distances are canonical neural computations," what examples from the neuroscience literature support this claim?
>
> We perform simultaneous stimuli discriminations in many modalities, similarly to normalization.
>
> kokB: Is it possible to identify real systems for which latent vulnerabilities exist? In my opinion the strength of the contribution would be dramatically increased if (1) it could be shown in a practical system that inducing robustness around point A leads to decreased robustness around point B, and (2) that the perceptual Ricci curvature could be used to identify the risk of this type of vulnerability?
>
> We appreciate this excellent question: Latent vulnerabilities exist only at locations which cannot be attacked by adversarial Doppelgangers. The theoretical necessity for certified adversarial robustness with a perceptual radius of $d_{\infty} \geq 1$ is currently unmet by any SOTA classifier, as the reviewer implies. The paper provides a warning about new risks that will arise when we finally achieve sufficient robustness.

---

### Official Review · Reviewer_o38w · 2025-11-03

**Soundness:** 1
**Presentation:** 2
**Contribution:** 1
**Rating:** 0
**Confidence:** 2

**Summary:**

This paper claims to formalize the coherence in classification. The paper first provides the background notions, including the indiscriminability relation, perceptual distance, and conceptual accuracy. The paper provides a single example that measuring perceptual distance is feasible. The paper also attempts to argue that adversarially robust classifiers cannot be achieved due to the lack of unambiguous concepts in classification problems. Rather than pursuing the adversarial robust classifier, the paper claims that we can instead achieve a low coherency failure rate in practice. Then, the paper shifts the focus to perceptual generalization and latent adversarial vulnerability. The paper explains the latent adversarial vulnerability with the perceptual drift, then defines the perceptual curvature that can be used to identify the perceptual drift.

**Strengths:**

I don’t see a clear strength in this paper. At least, the paper is trying to do something new.

**Weaknesses:**

1. The paper’s motivation is unclear. The paper provides several definitions, the realization of which is not even guaranteed. Not only do people lack the ability to quantify the proposed definitions, but it is also unclear whether computing (or approximating) the quantities is tractable. If all the concepts cannot be implemented in practice, I don’t see a reason why the machine learning community should care about those notions at all.
2. The paper writing is generally bad. The topic of discussion changes without enough context, making it difficult for the reader to follow what the paper is trying to say.
3. The paper’s contents are largely speculative, lacking sufficient support or evidence. The paper provides at most one or two examples of the proposed concepts, which is clearly not enough to demonstrate anything.
4. Even the reference paper on which this paper is grounded has the same problem. The reference paper presents only one example of an “adversarial doppelganger” whose background is extremely dark (which made it harder to recognize whatever perturbation in that region), and there is no experimental support that this unrecognizable perturbation is common. The reference paper is also speculative, as it contains only speculations that cannot be experimentally verified.
5. If I understand correctly, the indiscriminability relation depends on a “specific subject”. (The reference paper at least mentions this, but this paper does not clearly say this.) Therefore, all the derived concepts, such as perceptual topology, degrees of separation, and perceptually coherent class partition, depend on the subject and are not well-defined. Measuring a classifier’s conceptual accuracy by measuring the mismatch does not make sense because there is no single notion of perceptual coherence, and it depends on the subject that is used to define the perceptual coherence. The author should clarify how a unified measure of conceptual accuracy can be established across different subjects.
6. The network presented in Example 0 is specific to a single example, and there is no guarantee that it generalizes to real-world examples. A more concrete example that reflects a real-world setting should be provided.
7. Example 2 is not proper because the class label was not classified by a model trained to classify such an image. While CoPilot may have an image classification model that aided its answer, we have no information about the underlying model, and it is possible that CoPilot utilized different models for each query.
8. Example 3 just contains a specific choice that fits the proposed conceptual coherence. However, there is no evidence that the perceptual neighborhood is generally measurable, so the proposed coherency failure rate could be useless in practice.
9. The Example of the latent adversarial vulnerability phenomenon is also hypothetical, and there is no evidence that the attacker can intentionally induce such a phenomenon. In other words, there is no clear evidence that the attacker can realize the proposed Ouroboros attacks. If it is a realizable concept, then the paper should provide a prototype implementation.
10. The proposed concept of perceptual curvature has no practical value. In Section 4, the paper says, “However, to date, we do not know how to ‘code’ these computations.” Why should the adversarial machine learning community care about some concept whose realization is unclear?

**Questions:**

1. Choose a clearer title that directly describes what this paper is about.
2. If I understand correctly, many proposed notions should depend on the subject who is perceiving the context. If my understanding is incorrect, please justify why the perception is not dependent on the subject. If the perception depends on the subject, specify the perceptual context to which the paper refers.
3. Provide empirical evidence that the proposed concepts are common in the machine learning context. I’m not asking for one or two examples that fit the proposed definition, or literature that does not consider machine learning practice. If there is no such evidence, justify the reason why the machine learning community should care.
4. Design and perform more systematic experiments. As pointed out in the Weaknesses, querying CoPilot to classify three images is not a well-designed experiment.
5. At least, I agree on one statement. “Rather than searching for high-accuracy robust classifiers, …, it is meaningful to identify classifiers that achieve high accuracy while maintaining adversarial vulnerability below the maximal tolerable threshold.” In my opinion, the adversarial machine learning community implicitly knows this, but we just pursue the former because it is more cost-effective. Attempting to demonstrate that the latter would yield a better result is a commendable effort; however, I disagree that this paper effectively makes the point.

---

> ### Author Response · Authors · 2025-12-04
> **Factual Corrections:**
>
> o38w: [Doppelgangers] and “paper’s contents are largely speculative”
>
> The existence of perceptually indistinguishable stimuli – Doppelgangers – is a well-known phenomenon it has been studied and exploited for thousands of years, e.g., they have been using to build Sorites chains, investigated in psychophysics (JND), cognitive linguistics, colorimetry, and epistemology. There are thousands of adversarial Doppelganger examples produced in computer vision – the adversarial examples that made Szegedy et al.,  " Intriguing properties of neural networks" seminal are precisely adversarial Doppelgängers.  Similarly, the idea that the concepts and categories are often ambiguous is a staple in cognitive science and linguistics. These are not speculations but well-established phenomena. This paper and the “reference” paper provide the machinery to provide rigorous reasoning and computational frameworks to compute in the ambiguous perceptual spaces.
>
> o38w: “If I understand correctly, the indiscriminability relation depends on a “specific subject”..."
>
> The “subject” in the definition of indiscriminability is the “standard observer” a concept that is well known in psychology, psychophysics, etc.
>
> 038w: “The network presented in Example 0 is specific to a single example, and there is no guarantee that it generalizes to real-world examples….”
>
> As stated in the paper this network is the canonical neural computation derived from Weber's Law, it is generalizable across all real sensory modalities that satisfy the law (hue, tone, brightness, etc.).
>
> o38w: “Example 2 is not proper because the class label was not classified by a model trained to classify such an image…”
>
> The model's identity is irrelevant. As explained in every classifier will fail along the random walk detailed in Example 1. We will provide the code to generate these random walks.
>
> 038w: “Example 3 just contains a specific choice that fits the proposed conceptual coherence. However, there is no evidence that the perceptual neighborhood is generally measurable, so the proposed coherency failure rate could be useless in practice.”
>
> The example is not an abstract construction it comes from colorimetry and acoustics. The measure of a perceptual neighborhood is exactly the probability to encounter a phenomenon.

---

### Note · Authors · 2026-01-20

**Comment:**

We are withdrawing Submission 949. This decision is not based on a disagreement over scientific merit, but on a demonstrable breakdown of the peer-review process. The following points summarize some of the most critical factual and procedural errors in the reviews:

1.  **Verification of Procedural Failure (Supplementary Materials) **

Two reviewers (zoRc, ZYBz) justified their low scores by claiming a lack of supplementary materials, algorithms, and empirical tractability.

+ The Fact: The supplementary zip file (949_Suppl.zip) was successfully uploaded and has been available on OpenReview throughout the process.
+ The Content: These materials contain the exact "missing" items: a labeled dataset, an actionable program to compute conceptual margins, and a worked-out example.


2.  **Some Fundamental Factual Misunderstandings **
+  **Adversarial Doppelgängers **: Reviewers labeled the existence of "Doppelgängers" (perceptually indiscriminable stimuli) as "speculative." In reality, the Adversarial Examples established by Szegedy et al. (2013)—the very phenomenon that founded this field of research—are Adversarial Doppelgangers.
+  **Concrete Algorithms **: Reviewers claimed a lack of concrete algorithms ignoring the canonical neural circuit for computing the perceptual distance $d_{\infty}$ which is important because it applies for a wide and foundational class of stimuli provided in the paper; the explicit report of the Conceptual Margin ($\mathcal{M}=3$) in the Supplementary Materials.
+  **Quantitative Measurability **: Reviewers expressed doubts that it is possible to compute the measure of the perceptual neighborhoods. However, this is never the case in real applications in which this measure is precisely the probability to encounter the phenomena corresponding to the stimuli (e.g., the probability to encounter certain shade of green in a psychophysics experiment).

**Withdrawal Confirmation:**

I have read and agree with the venue's withdrawal policy on behalf of myself and my co-authors.